# Dynamic heterogeneity in COVID-19: Insights from a mathematical model

Chrysovalantis Voutouri[1,2☯], C. Corey Hardin[3☯], Vivek Naranbhai[4,5,6], Mohammad R. Nikmaneshi[1], Melin J. Khandekar[7], Justin F. Gainor[4], Lance L. Munn[1]*, Rakesh K. Jain[1]*, Triantafyllos Stylianopoulos[2]*

1 Department of Radiation Oncology, Edwin L Steele Laboratories, Massachusetts General Hospital and Harvard Medical School, Boston, MA, United States of America, 2 Department of Mechanical and Manufacturing Engineering, Cancer Biophysics Laboratory, University of Cyprus, Nicosia, Cyprus, 3 Department of Pulmonary and Critical Care Medicine, Massachusetts General Hospital and Harvard Medical School, Boston, MA, United States of America, 4 Department of Medicine, Massachusetts General Hospital Cancer Center, Division of Hematology/Oncology, Massachusetts General Hospital, Boston, MA, United States of America, 5 Dana-Farber Cancer Institute, Boston, MA, United States of America, 6 Center for the AIDS Programme of Research in South Africa, Durban, South Africa, 7 Department of Radiation Oncology, Massachusetts General Hospital and Harvard Medical School, Boston, MA, United States of America

☯ These authors contributed equally to this work.
* lmunn@mgh.harvard.edu (LLM); rjain@mgh.harvard.edu (RKJ); tstylian@ucy.ac.cy (TS)

**Data Availability Statement:** All data supporting the findings of this study are available in the paper and the Supplementary Information. The COMSOL

## Abstract

Critical illness, such as severe COVID-19, is heterogenous in presentation and treatment response. However, it remains possible that clinical course may be influenced by dynamic and/or random events such that similar patients subject to similar injuries may yet follow different trajectories. We deployed a mechanistic mathematical model of COVID-19 to determine the range of possible clinical courses after SARS-CoV-2 infection, which may follow from specific changes in viral properties, immune properties, treatment modality and random external factors such as initial viral load. We find that treatment efficacy and baseline patient or viral features are not the sole determinant of outcome. We found patients with enhanced innate or adaptive immune responses can experience poor viral control, resolution of infection or non-infectious inflammatory injury depending on treatment efficacy and initial viral load. Hypoxemia may result from poor viral control or ongoing inflammation despite effective viral control. Adaptive immune responses may be inhibited by very early effective therapy, resulting in viral load rebound after cessation of therapy. Our model suggests individual disease course may be influenced by the interaction between external and patient-intrinsic factors. These data have implications for the reproducibility of clinical trial cohorts and timing of optimal treatment.

## Introduction

Severe COVID-19, like the Acute Respiratory Distress Syndrome (ARDS) from other causes, is highly heterogeneous. Presentations range from rapid resolution [1] to persistent organ failure

code is available at Zenodo: https://doi.org/10.5281/zenodo.10995291.

**Funding:** This work was supported by Rakesh Jain's research is supported by R01-CA259253, R01-CA208205, R01-NS118929, U01-CA261842, and U01-CA 224348, Outstanding Investigator Award R35-CA197743 and grants from the National Foundation for Cancer Research, Jane's Trust Foundation, Niles Albright Research Foundation and Harvard Ludwig Cancer Center. Lance Munn's research is supported by R01-CA2044949. Triantafyllos Stylianopoulos's research is supported by the European Research Council ERC-2019-CoG-863955. Chrysovalantis Voutouri is supported by Marie Skłodowska Curie Actions Individual Fellowship Global Horizon 2020 MSCA-IF-GF-2020-101028945. The funders had no role in study design, data collection and analysis, decision to publish, or preparation of the manuscript.

**Competing interests:** JFG has served as a compensated consultant or received honoraria from Bristol-Myers Squibb, Genentech/Roche, Ariad/Takeda, Loxo/Lilly, Blueprint, Oncorus, Regeneron, Gilead, Moderna, Mirati, AstraZeneca, Pfizer, Novartis, iTeos, Nuvalent, Karyopharm, Beigene, Silverback Therapeutics, Merck, and GlydeBio; research support from Novartis, Genentech/Roche, and Ariad/Takeda; institutional research support from Bristol-Myers Squibb, Tesaro, Moderna, Blueprint, Jounce, Array Biopharma, Merck, Adaptimmune, Novartis, and Alexo; and has an immediate family member who is an employee with equity at Ironwood Pharmaceuticals. LLM owns equity in Bayer AG and is a consultant for SimBiosys. RKJ received consultant fees from Elpis, Innocoll, SPARC, SynDevRx; owns equity in Accurius, Enlight, Ophthotech, SynDevRx; and serves on the Boards of Trustees of Tekla Healthcare Investors, Tekla Life Sciences Investors, Tekla Healthcare Opportunities Fund, Tekla World Healthcare Fund; and received a grant from Boehringer Ingelheim. Neither any reagent nor any funding from these organizations was used in this study. Other co-authors have no conflict of interests to declare. This does not alter our adherence to PLOS ONE policies on sharing data and materials.

and death [2]. Heterogenous clinical courses may arise due to diverse features of the virus, as in the emergence of viral variants, or varied patient comorbidities. Severe disease following COVID-19 infection however, involves a complex set of interactions between viral replication, innate [3] and adaptive immune responses [4], patient comorbidities [5], and efficacy of therapeutic agents. Complex interactions such as this may give rise dynamic heterogeneity, in which varied initial conditions lead to varied outcome in similar patients and to unexpected phenomena such as 'rebound' of SARS-CoV-2 infectivity after cessation of antiviral treatment [6]. Isolating dynamic phenomenon such as these in viral infection and sepsis [7] is difficult, if not impossible, with clinical observations alone as key events, such as size of initial infectious inoculum, may have occurred prior to presentation or even prior to onset of symptoms. Mechanistic mathematical modeling [8–11] can be used to explore a wider range of scenarios than is practical to test in patients and may thus help identify novel determents of COVID-19 clinical course.

We have developed a mechanistic model of COVID-19 that incorporates the major aspects of pathogenesis [12, 13]. In this study, we deploy our model to study how the possible trajectories of viral load and arterial oxygen saturation following infection are affected by changes in infectious load, changes in the potency of antiviral agents, changes in the magnitude of the innate immune response and coagulation cascade and by changes to the adaptive immune response, including immunization and the presence of viral variants with varying degrees of immune evasion. We find that the complex dynamics of disease progression allows for clinical courses in which similar patients go on to experience divergent outcomes. In particular, we find that clinical course may be substantially modified not only by vaccination, treatment efficacy and inherent patient or pathogen characteristics, but also by external factors such as the initial viral load. Further, we find an interaction between time of treatment onset and the robustness of the immune response which may explain viral rebound [14]. These findings have implications for understanding heterogeneity in critical illness and suggest that clinical heterogeneity may arise from both modifiable and non-modifiable sources.

## Materials and methods

### Description of mathematical model

We developed a mechanistic model of COVID-19 infection which consists of a series of linear and non-linear differential equations which describe the dynamics of infection of epithelial cells in the lung by SARS-CoV-2, the innate immune response to infection, including the production of pro- and anti-inflammatory cytokines and the activation of the coagulation cascade. The model has been extensively described previously [13, 15, 16] The model further accounts for interactions between the virus and immune cells including neutrophils, B cells and T cells (**Fig 1** and **S1 Fig** in **S1 Appendix**). A detailed description of the equations and parameters can be found in prior publications and the Supporting Information.

This model encompasses intricate interactions occurring within the human body, utilizing differential equations to simulate various aspects of SARS-CoV-2 infection and the corresponding immune responses. These include the viral entry process, immune system activation, cytokine production, and the coagulation cascade. Furthermore, our model incorporates mechanisms and immune responses related to both mRNA and vector-based vaccines. Of particular importance is the inclusion of a pharmacokinetic-pharmacodynamic model, which meticulously tracks the movement of viral particles and other relevant elements across major body compartments. This comprehensive framework empowers us to conduct in-depth analyses and make predictions.

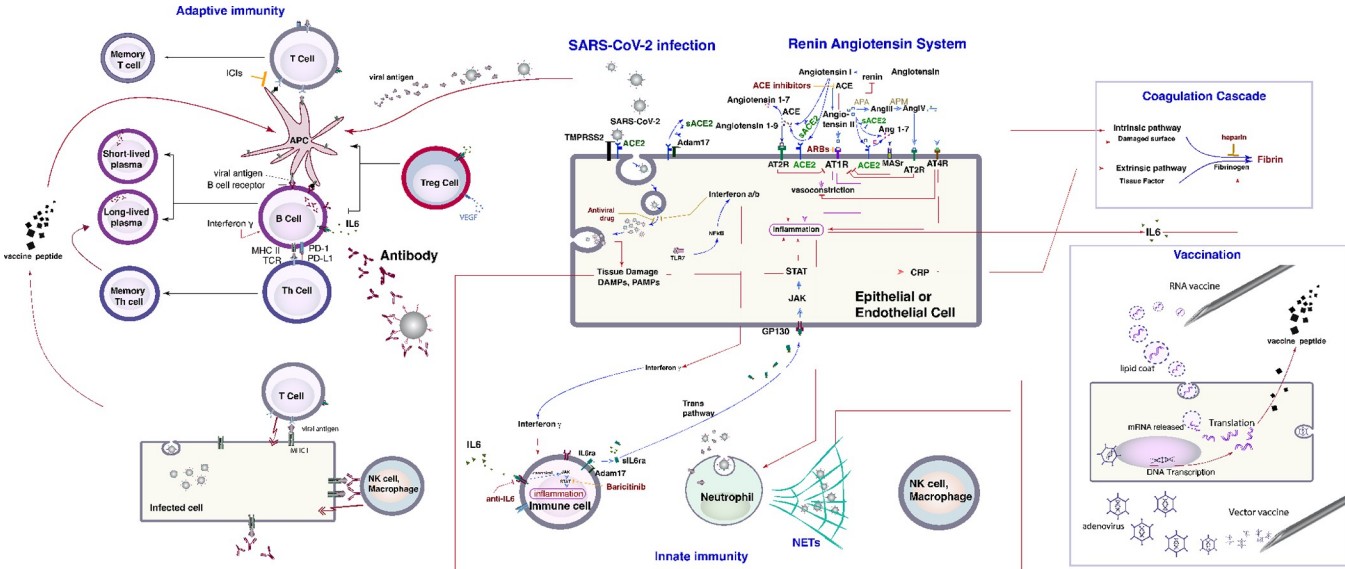

**Fig 1. Biological interplay and pathways incorporated by the mathematical model.** The basic components of the model are: i) a detailed model of lung infection by SARS-CoV-2 that includes innate and adaptive immune responses, known mechanisms of the renin-angiotensin system (RAS) and the coagulation cascade. Intracellular virus initiates inflammatory pathways through toll-like receptors and NFκB, which produces interferons and other inflammatory cytokines. Antiviral drugs affect the replication of the virus within host cells. The viral antigens, along with inflammatory cytokines, facilitate activation of naïve B and T cells, creating virus-specific effector cells. Activation of naïve immune cells is controlled by viral antigen strength and the status of immune checkpoint inhibition (specifically PD-L1/PD-1). Ii) A Pharmacokinetic/Pharmacodynamic model of dissemination of viral particles, cytokines, micro-thrombi and antibodies in the major organs (lung, heart, liver, brain, spleen, gastro-intestinal, upper body, lower body, torso, cardiac vessels and the tumor). Iii) All steps of vaccination-induced immunity for mRNA and vector vaccines, including the translation of viral antigens, the production of antigen presenting cells by dendritic cells, the subsequent activation of T cells and B cells to create CD4[+] and CD8[+] effector and memory T cells as well as short-lived and long-lived plasma (antibody-secreting) B cells. Iv) Tumor cells and interactions with the immune system. Proliferation of tumor cells depends on oxygen levels in the tissue, and their death rate on the interaction of cancer cells with immune cells (effector CD8[+] T cells, natural killer cells, type 1 macrophages and neutrophils) as well as on the effect of cancer therapy. This schematic Illustrates the foundational elements of our model, including the detailed lung infection mechanism by SARS-CoV-2 and the innate and adaptive immune responses. It highlights key processes such as the activation of immune pathways by the virus, effects of antiviral drugs, and the role of immune checkpoints. The schematic also shows the model's representation of the pharmacokinetic/pharmacodynamic dissemination of viral particles and micro-thrombi across major organs, as well as the distinct steps of vaccination-induced immunity for mRNA and vector vaccines. Lastly, it depicts the interaction between tumor cells and the immune system, considering factors like oxygen levels and the impact of cancer therapy.

## Modeling strategy and model formulation

Our model contains a large number of parameters, many of which may not be explicitly determined from existing experimental data. Even where such experimental data exists, the large number of parameters results in a danger of overfitting. Due to this limitation, we do not seek to determine optimum parameter values for predicting the disease trajectory of any individual patient. Instead, here we seek to understand the full scope of possible clinical courses which follow SAR-CoV-2 infection and the factors which influence them. To do so, we study the range of possible outcomes which correspond to a range of possible parameter values. In order to understand the complex interaction between viral characteristics, innate and adaptive immunity, random events and treatment efficacy, we explicitly include terms describing the response to vaccination, varying levels of antibody affinity to viral variants and anti-viral drugs of varying efficacy. We model an individual who has been vaccinated against the ancestral strain of SARS-CoV-2 with the BNT-162b2a mRNA vaccine, including a first booster dose. Viral infection is assumed to take place 6 months following the booster dose (the model allows for the variation of antibody level over time [12].

The initial values of the model parameters related to viral infection and the pharmacokinetics/pharmacodynamics of COVID-19 were defined in previous work. Also, model parameters

related to vaccination-induced immunity, including the affinity for and uptake rate of vaccine particles by cells, the rate of DNA transcription to mRNA, the production rate of viral antigens and the degradation rates of the vaccine and the viral antigen have been reported previously [12]. The values of model parameters are summarized in **S1 Table in S1 Appendix**.

From studies of neutralization experiments that involved all omicron variants from BA.1 to BA 4/5 [17–20], we determine the affinity of the vaccine-induced antibodies for each SARS-CoV-2 variant (**S2 Fig in S1 Appendix**) [17–20]. We then simulate infection and treatment in patients who received the BNT-162b2a initial vaccine and a first booster, and then were infected with the ancestral strain or an omicron variant.

## Incorporation of antiviral drugs and monoclonal antibodies

To incorporate antiviral treatment into the model, we assumed that antiviral drugs reduce the rate of release of replicated virus, $K_{in}$. Furthermore, to contextualize the extent of decrease in virus replication for the most common antivirals, namely remdesivir, molnupiravir and nirmatrelvir+ritonavir (NMV/r), we employed data from clinical studies for the proportion of patients that recovered from COVID-19 having been treated with these drugs [21–23]. We calculated the baseline value of $K_{in}$ for our studies by fitting the model to these clinical data, having $K_{in}$ as the only fitting parameter. Comparison of model predictions with clinical data for the three antivirals along with the corresponding values of $K_{in}$ are shown in **S3 Fig in S1 Appendix**. In addition, to assess the sensitivity of our predictions to variations in other model parameter values, we repeated simulations while altering parameters other than $K_{in}$ using a range of values within an order of magnitude around the baseline values (**S2 Table in S1 Appendix**). Simulations were repeated for all possible combinations among parameters taking 100 different values for each parameter. In this way we attempt to quantify the effect of uncertainty in model parameter values.

The results are presented in the figures as standard error bars from the baseline values. The model was able to provide accurate predictions of the proportion of patients recovered with $\chi^2$ lower than 0.0123 in all cases. Monoclonal antibodies were treated in the model separately from antibodies produced by vaccination or viral infection, so that three types of antibodies exist: (antibodies produced by vaccination, those produced by infection and monoclonal antibodies) which may vary in their affinity and half-life. Initial parameter values related to monoclonal antibodies were obtained by fitting the model to clinical data of patients who recovered from COVID-19 after receiving these drugs. Comparison of model predictions with the clinical data and the values of relevant parameters for the bamlanivimab and etesevimab antibodies and for the casirivimab and imdevimab antibodies are shown in **S4 Fig in S1 Appendix**. Similar to the antivirals, the model was able to provide accurate predictions of the proportion of patients recovered ($\chi^2 < 0.0236$).

## Results

### Interaction of antiviral therapy with heterogeneity in vaccine induced immunity

Clinical data suggest that prior vaccination may modify the utility of antiviral therapy [24]. To better understand this interaction, we first assessed how various antiviral therapies affect viral replication and oxygen saturation when administered in the setting of prior vaccination and with infection by viral variants with varying degrees of immune evasion (i.e., virus-antibody affinity; **Fig 2**). Panels 1A,B confirm that antiviral therapy confers benefit even in the setting of vaccination and that effective antiviral therapy is more important to outcome in the setting of

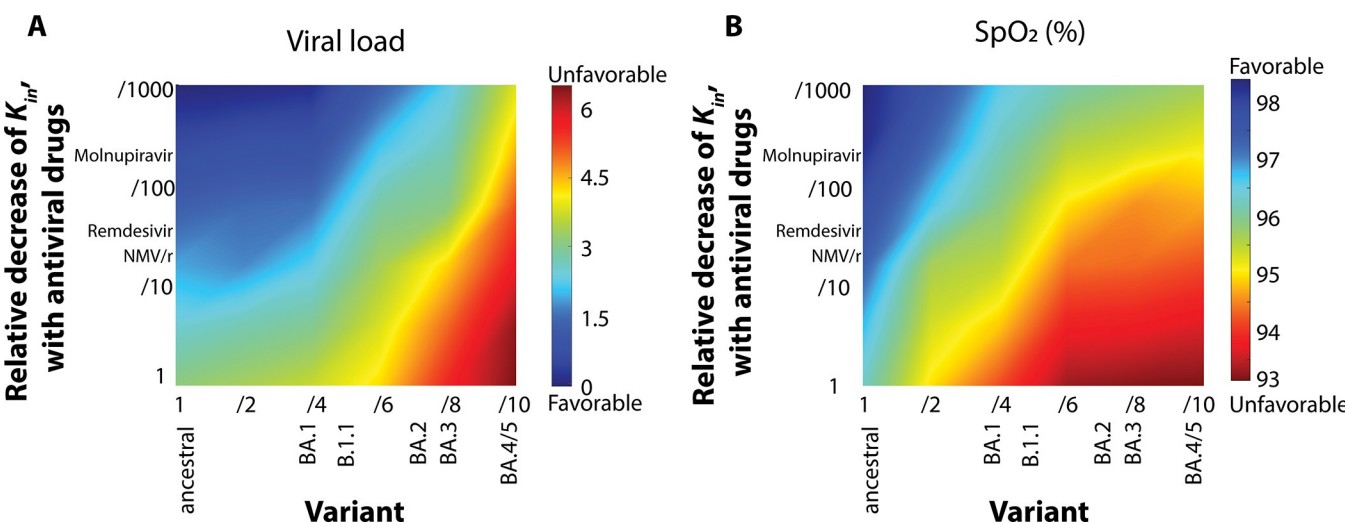

**Fig 2. Antiviral therapies and viral strain affect COVID-19 severity.** Diagrams of peak viral load (A) and minimum oxygen saturation (SpO$_2$) (B) caused by viral infection as a function of viral immune evasion and relative decrease of virus replication rate, K$_{in}$, with antiviral drugs. 'Favorable' outcomes for viral load correspond to low concentrations, while 'unfavorable' outcomes relate to high concentrations. For oxygen saturation, 'favorable' outcomes are indicated by high levels, and 'unfavorable' outcomes by low levels. K$_{in}$ values estimated to correspond to current antivirals are shown in the plot. Fold-decrease in antibody affinity for the virus is on horizontal axis, and values thought to correspond to currently known variants (all omicron variants from BA.1 to BA 4/5 are listed for reference. Viral load is normalized to the initial value.

variants which evade vaccine induced immunity. Our model suggests that for monoclonal antibodies to be effective against the BA4/5 variant, they need to not only have a high affinity to the virus but also exhibit blood circulation times longer than 50 days.

In our study, key control parameters that impact COVID-19 outcomes include the basic reproduction number, antiviral treatment efficacy, immune response rates, and initial viral load, which all influence epidemic peaks and virus spread. Variations in treatment efficacy markedly affect disease progression and population spread dynamics. Differing immune responses lead to varied disease severities and recovery rates, highlighting individual immune variability. The initial viral load also plays a vital role, with higher loads linked to more severe disease courses. Understanding these parameters is essential for applying our model to real-world scenarios.

In setting the model's parameter ranges, we grounded our choices in physiological and experimental evidence. For instance, the range for the basic reproduction number (R0) was based on published studies on COVID-19 transmission rates. The efficacy of antiviral treatments was modeled based on clinical trial data, reflecting both current and potential future therapies. Immune response parameters were informed by immunological studies, acknowledging the diversity in individual responses to the virus. Finally, the initial viral load parameters were derived from virological research, recognizing the variability in viral exposure and infection dynamics. This evidence-based approach ensures that our model's simulations are both realistic and relevant to the diverse scenarios observed in the pandemic.

## Interaction of antiviral therapy with heterogeneity in the innate and adaptive immune response

Previous work has highlighted the importance of the adaptive, T-cell response to illness resolution [13]. Moreover, COVID-19 patients may vary in degree of hypercoagulability and in the amplitude of either the innate or adaptive response to a given stimulus [25]. To better

understand the interaction of these diverse pathophysiologic mechanisms, we simulated the clinical course of antiviral treatment after imposing various perturbations to the immune response and coagulation cascade. We tested three cases: patients with i) varying levels of CD4 and CD8 immune cell activation, ii) varying levels of pro-inflammatory cytokine production, and iii) varying tendency for thrombus formation in response to infection and immune activation. The values of model parameters modulated to account for these three cases are shown in S1 Table **in S1 Appendix**.

In the analysis, we changed the indicated parameter values up to 1000-fold. We also focused on vaccinated patients, using a level of antibody response consistent with the BA4/5 variant. Results are presented in **Fig 3**. Predictably, we find that high rates of immune cell activation and inflammatory cytokine production result in lower viral load–and that viral control is augmented by effective antiviral therapy (moving from bottom to top in all panels in **Fig 3**). However, we also find that when pro-inflammatory cytokine production is increased, the decreased viral load comes at the cost of poorer gas exchange (**Fig 3C**, moving from left to right). Increasing the effectiveness of anti-viral therapy in the setting of increased inflammatory cytokine production attenuates the adverse effects on gas exchange (consistent with the more rapid decrease in inflammatory stimulus). Interestingly, increases in the CD4 and CD8 T-cell response augment viral control with less adverse effect on gas exchange. Imposing changes in microthrombosis production rate independent of cytokine production, we see that enhanced thrombosis also adversely affects oxygenation (moving from left to right in **Fig 3C**). In **S4 Fig in S1 Appendix** we show the combined effect of hyperinflammation and hypercoagulability in which the effect on gas exchange is increased when compared to altering either process alone.

We next sought to understand more about the mechanisms that underly the above findings. **S5A Fig in S1 Appendix** indicates that increasing the rate of production of proinflammatory cytokines enhances recruitment of both macrophages and neutrophils to the lung, while increasingly-effective antiviral therapy decreases both.

To further understand the role of these innate immune effector cells in controlling viral load in response to increased cytokine production, we also investigated the case where we mathematically deleted the neutrophil population (**S5B Fig in S1 Appendix**). In this case, we still observe worsened oxygen saturation with elevated levels of cytokine production, as in the case with neutrophils present (**Fig 2C**, right). This is associated with increased burden of microthrombus (**S6 Fig in S1 Appendix**), which limits blood perfusion and oxygen transport. However, without neutrophils, this apparent inflammatory injury is not associated with the same degree of viral control(compare **S6C and S7B Figs in S1 Appendix**). Augmenting the antibody response via increasing B cells results in both decreased peak viral load and increased nadir SpO2 (**S6 Fig in S1 Appendix**).

## Interaction of initial viral load with treatment efficacy and immune response

An additional potential source of heterogeneity in clinical course are stochastic environmental factors, such as initial viral inoculum. In **Fig 4**, we examine the peak viral load and nadir SpO2 that results from varying the magnitude of the initial viral exposure. Increasing viral inoculum is associated with worsening SpO2 and higher peak viral load at all levels of anti-viral efficacy and pre-existing immunity.

We further examined the interaction between heterogeneity in the inflammatory response and in the initial viral inoculum, focusing again on the level of antibody response consistent with the highest risk BA.4/5 variant. In **Fig 5,** we demonstrate the higher peak viral load associated with higher initial viral load is attenuated by a more aggressive inflammatory response.

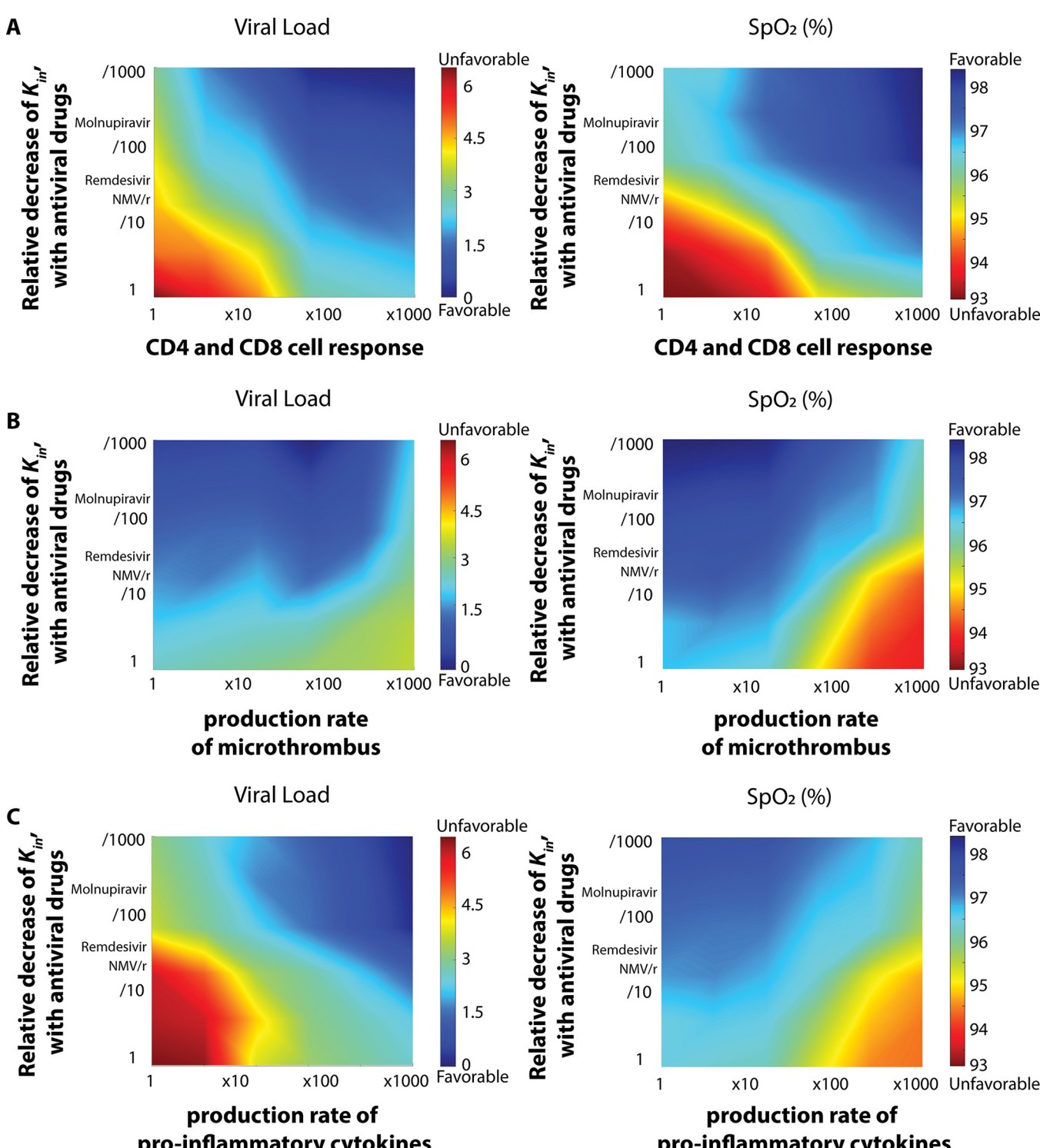

**Fig 3. Heterogeneity in the immune response affects viral clearance and oxygen saturation.** Diagrams of peak viral load with varying decreases in in viral replication (Kin) as a function of CD4 and CD8 cell response, production of pro-inflammatory cytokines and production rates of microthrombus. Viral load is normalized by division with the initial value.

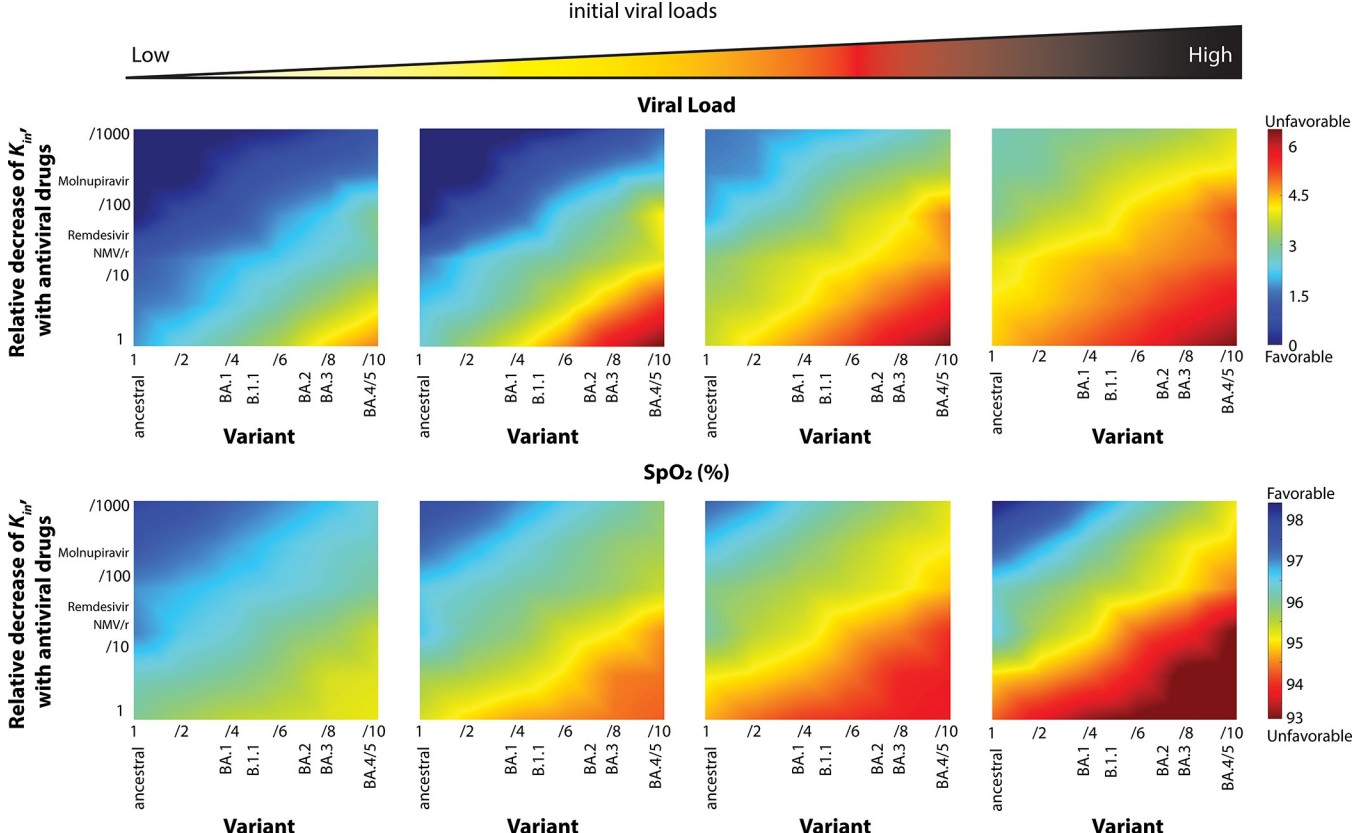

**Fig 4. Increasing viral inoculum compromises immune and treatment efficacy.** Diagrams of peak viral load and nadir SpO2 at varying levels of effectiveness of vaccine-induced antibodies and antiviral therapy. Initial viral load increases from left to right and in all cases is associated with worsening clinical course (at any given level of pre-existing immunity or antiviral efficacy). Viral load is normalized by division with the initial value $4.96 \times 10^4$ [pg/ml]. The low and high initial viral load are set at $4.96 \times 10^2$ [pg/ml] and $4.96 \times 10^6$ [pg/ml], respectively.

However, very high initial viral loads can result in high peak viral loads even at very high levels of expression of proinflammatory cytokines. The same is true of high levels of production of CD4 and CD8 T-cells. Similar dynamics are observed when looking at viral load and SpO2 over time in untreated patients (**S7 Fig in S1 Appendix**)

## Rebound after antiviral therapy and optimal treatment initiation

An unexpected clinical observation during antiviral drug therapy is the rebound of viral load after therapy termination, which has been attributed to interactions between innate and adaptive immune responses [26]. We varied the day of treatment initiation within a period of 10 days from infection. Here we define infection as first contact with virus and account for three different types of patients: a high risk (older) vaccinated patient, a low risk (young) vaccinated patient and a low risk (young) unvaccinated patient. The different parameter values employed to represent the immune state of the old and young individual is shown in **S4 Table in S1 Appendix**. Antiviral drug administration lasted for a period of 10 days from treatment initiation and the effects of treatment on the levels of infected host cells, SpO₂, microthrombosis in the lung and levels of activated CD8+ T cells for a high risk (older) vaccinated patient is presented in **Fig 6** for the case of treatment with NMV/r and **S8-S10 Figs in S1 Appendix** for NMV/r low risk vaccinated and unvaccinated patient, remdesivir and Molnupiravir.

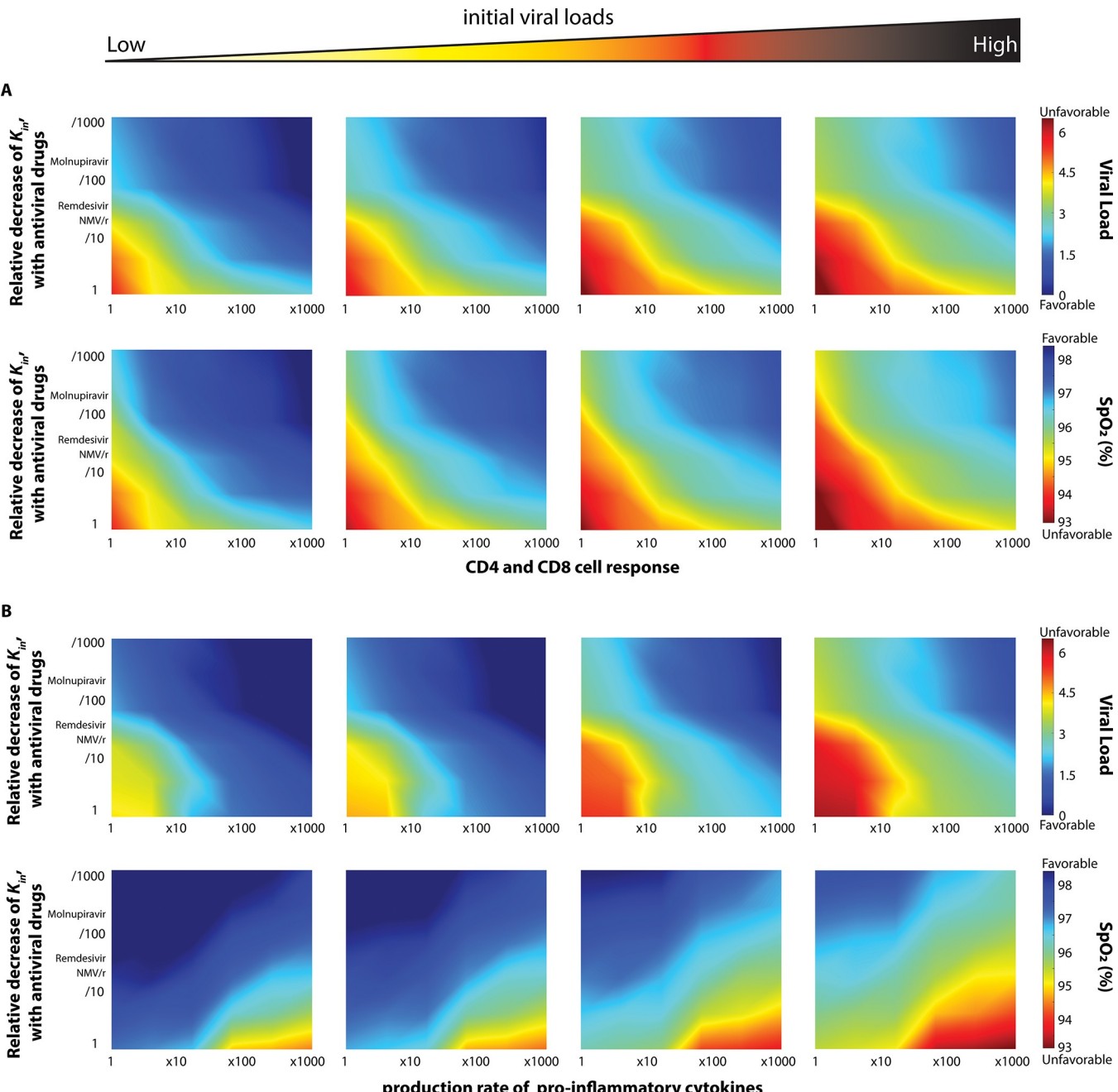

**Fig 5. Initial viral load determines disease severity (analysis of the BA4/5 variant).** The worsening clinical course (peak viral load and SpO2) associated with higher initial viral load is attenuated by more a more aggressive (increased rate of production of activated CD4 and CD8 cells or increased rate of expression of pro-inflammatory cytokines). Viral load is normalized by division with the initial value 4.96x10$^4$ [pg/ml]. The low and high initial viral loads are 4.96x10$^2$ [pg/ml] and 4.96x10$^6$ [pg/ml], respectively.

The model predicts that early treatment, before day 3 of infection, leads to the rebound of the disease after cessation of treatment, demonstrated by the increase in the levels of infected host cells. Whereas delayed treatment, after day 5, results in a number of infected cells similar to the case of no treatment. The model predicts an optimal window around day 5 corresponding to the initial, post-infection peak in the number of infected cells. Adding the antiviral at

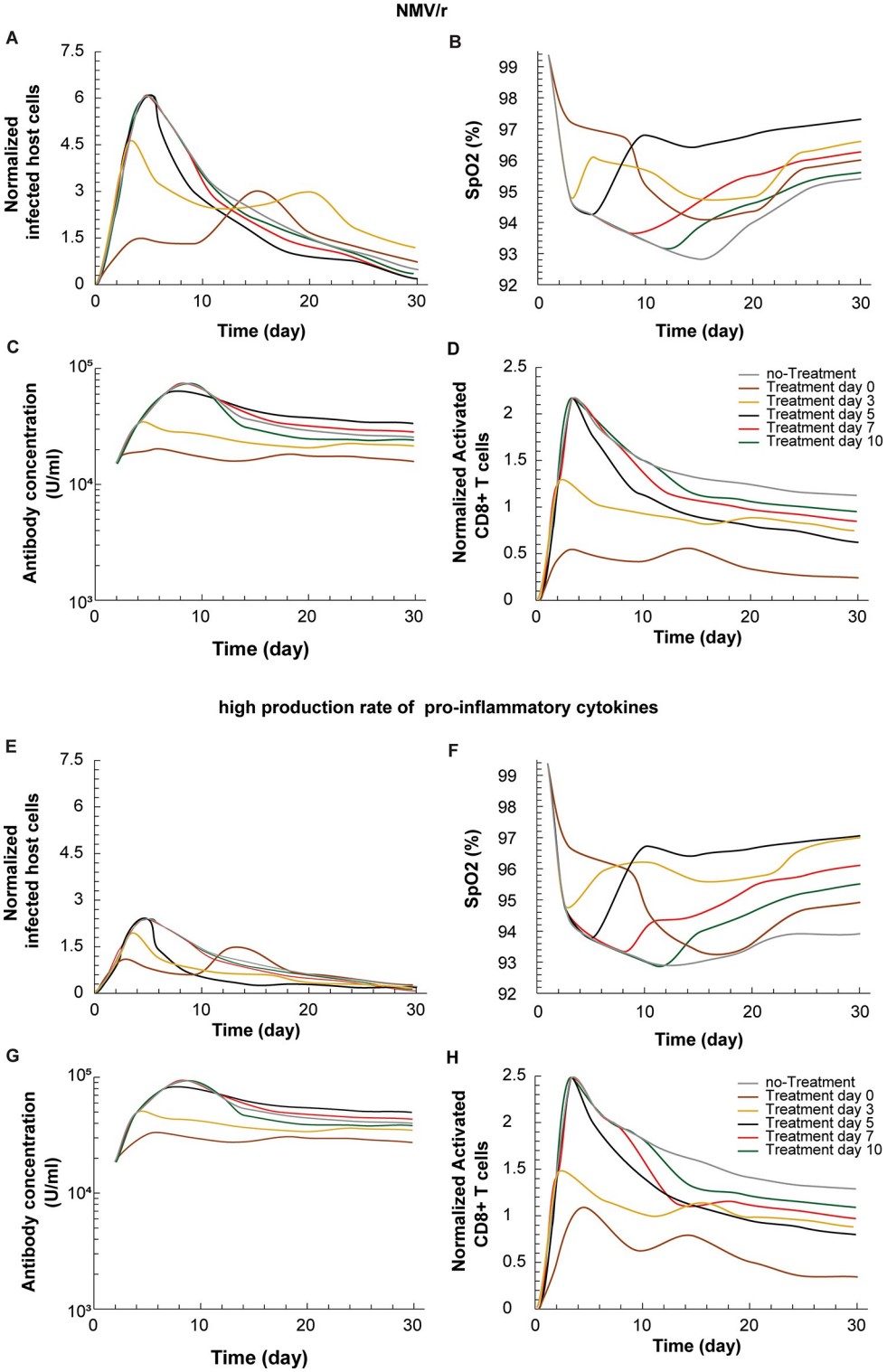

**Fig 6. Time of treatment initiation affects disease rebound and therapeutic outcome.** (A) Temporal variation in levels of infected host cells, (B) oxygen saturation, SpO$_2$, (C) levels of activated B cells and (D) levels of activated CD8$^+$ T cells for high risk (older) vaccinated induced by NMV/ras a function of treatment initiation. Antiviral treatments lasted for 10 days. Infected host cells and activated CD8$^+$ T cells are normalized by division with their initial values 14.46x10$^6$ [1/ml] and 0.4 [g/cm$^3$], respectively. (E) indicates that increasing the expression of proinflammatory

cytokines (by 1000-fold), decreases the peak number of infected cells with prolonged hypoxemia (F) the optimal timing for avoiding rebound or qualitatively effecting the degree of activated B cells (G) or activation of CD4 and CD8 T-cells (H).

this time results in the lowest number of infected cells at nadir and avoids rebound. Interestingly, this conclusion is valid for all types of patients and antiviral drugs, in contrast to previous reports [26]. The existence of rebound seems to be related to the dynamics of the adaptive immune response. Early treatment attenuates the adaptive response, and insufficient adaptive immunity develops (**Fig 6C& 6D**).

We next examined the combined effects of perturbations to the immune response on the timing of antiviral treatment and viral rebound (**Fig 6E and 6F**). With increases in the expression of pro-inflammatory cytokines, the peak number of infected cells is decreased but the optimal window for avoiding rebound remains unchanged.

## Discussion

A central challenge in the study of critical care syndromes such as sepsis and ARDS is the profound heterogeneity and complexity of the response to serious injury or infection. Heterogeneity may result from fundamental, qualitative biologic differences between patients but, as demonstrated here, may also arise from more subtle quantitative differences between patients. Complexity gives rise to heterogenous clinical trial results as stochastic factors can influence the treatment response in any given trial cohort [27, 28]. An advantage of mathematical modeling in this setting is the ability to precisely alter one aspect of pathophysiology and discern its effects on many others. Minimalist mathematical models [7, 11] have been used to identify multiple steady-states (resolution of infection, persistent infectious inflammation, persistent sterile inflammation) as a function of initial viral load and vigor/competence of the host immune response, but such minimalist models use lumped parameters that are not directly identifiable with clinically observable data. Here, we constructed a more detailed mechanistic model of COVID-19 which takes into account the major determinants of clinical course, including environmental (viral load), patient specific (magnitude of inflammatory response) and therapeutic parameters (antiviral agents and vaccine induced immunity). Here we illustrate the model's potential for assisting in the interpretation of clinical observations related to COVID-19 and outlined the validation procedure. The model's value is built on its comprehensive representation of viral dynamics and immune responses. It can project the course of the infection under varying scenarios, such as changes in viral strains or treatment strategies. For validation, we utilized a multi-step approach. Initially, the model was calibrated with current epidemiological data and clinical findings. Subsequently, its predictions were cross-verified with independent datasets, including emerging data on new viral variants and treatment responses.

We used this model to determine the interactions between antiviral therapies, initial infectious load and individual components of the innate and adaptive immune response. Reassuringly, the model confirms some expected relationships between viral characteristics, therapeutic efficacy, immune response and outcome. The model data, however, also suggest a number of novel hypotheses about the relationship between these multiple determinants of disease.

### Stochastic determinants of patient phenotype

In our model, the influence of random events on patient outcomes in COVID-19 is particularly significant, highlighting the consequences of disease complexity not just for COVID-19

but potentially a large number of respiratory infections. COVID-19 is characterized by a wide range of clinical manifestations, from asymptomatic cases to severe illness, not entirely explained by known risk factors. Our model incorporates these stochastic elements which as we highlight here, may constitute an underappreciated source of heterogenous clinical course. To the extent that initial conditions such as initial viral inoculum are difficult to control for in the setting of a clinical trial, such stochastic determinants of outcome may threaten the external validity of some trial cohorts. The ability of a trial to observe a positive effect from any single strategy may vary from trial to trial, depending on the unknown distribution of initial events Consistent with this possibility, latent class analysis has identified hypo and hyperinflammatory phenotypes of COVID-19 ARDS with frequencies that vary by cohort [29]

Our model thus demonstrates that severe illness is not always explainable by known risk factors. The model suggests that some combination of cytokine trajectories, viral load trajectory and patient comorbidities could be utilized as markers of optimal therapeutic strategy and as criteria to design optimal trial cohorts. This approach, while acknowledging the limitations in current data and understanding, underscores the potential of our model to contribute to the ongoing efforts in managing this dynamic and complex illness.

## Multivariate determinants of treatment response

In addition, our data highlight multiple potential determinants of clinical course. In **Fig 2**, it is clear that low viral load and normal gas exchange may result from either an effective antibody response or highly effective antiviral therapy [30] and the effects of each are reinforcing. This finding is consistent with clinical reports [30], providing confidence in the soundness of the model. At the same time, our results emphasize the quantitative nature of the relationship between host and pathogen—high degrees of immune escape limit the benefit from even improbably effective antiviral therapy. **Fig 3** illustrates that increasing proinflammatory cytokine expression can lead to lower viral load, but also lower oxygen saturation. This, too, is consistent with what is what is known about hyperinflammatory responses from the clinical literature, but the model data highlight the fact that multiple clinical courses are possible depending on the relative magnitudes of the various pathophysiologic and physiologic process. For example, the effect of hyperinflammation on oxygen saturation is exacerbated by hypercoagulability, a effect consistent with clinical experience in COVID-19 [31] but attenuated by increasing the CD4+ and CD8+ T cell response and by the provision of effective antiviral therapy. As such, each patient follows a trajectory through a high dimensional landscape with multiple routes to a given outcome.

## Inflammatory and infectious injury

Consistent with the findings of more minimal models [7] our data point to the existence of diverse degrees of infectious and inflammatory injury with potential treatment implications. Increases in pro-inflammatory cytokine production result in lower peak viral load but also lower nadir oxygen saturation consistent with a primarily inflammatory lung injury for these patients. This is most evident in the setting of highly effective antiviral therapy and maximal inflammatory cytokine expression (**Fig 3C**, right upper quadrant) in which peak viral load is very low but there is, nevertheless, decreased oxygen saturation. Interestingly, zeroing out the neutrophil response results in higher viral load even with maximal cytokine expression suggesting the possibility of both persistent infectious and inflammatory injury in the setting of neutropenia. This trajectory is consistent with some pre-clinical models of viral infection in which lack of neutrophils, despite their unclear role in direct viral clearance, worsens outcome [32]. For any given level of cytokine expression, viral load decreases and SpO2 increases with

increasingly effective antiviral therapy (i.e., decreased production of intact virions) highlighting the contribution of infectious injury even in the setting of hyperinflammation (**Fig 3C**, left lower quadrant). Interestingly, in our model the effect of uncontrolled viral load on SpO2 appears somewhat less than that of uncontrolled inflammation (compare the right and left ends of the lowest row of **Fig 3C**). To the extent that the model predicts hypoxemia with low viral load in the setting of elevated inflammatory cytokine production it is possible to hypothesize that some patients (those with more vigorous innate immune responses) will be more prone to inflammatory injury. Note, however, that the model does not specify the cause of increased inflammatory cytokine production. Even at the highest simulated level of proinflammatory cytokine production nadir SpO2 is increased with more effective antiviral therapy–suggesting that rapid pathogen control is an effective way to avoid respiratory failure even in patients prone to inflammatory injury. That this argument could extend to the utility of vaccination is suggested by the salutary effects on both viral load and SpO2 that result from augmenting the adaptive immune response in the model.

## Viral rebound and dynamic determination of treatment efficacy

Finally, we find an additional, quantitative interaction between adaptive immunity and viral load wherein rapid viral clearance in response to effective therapy, and the subsequent failure to generate a robust adaptive response, can lead to viral rebound after the cessation of therapy. As a result the model suggests a fairly narrow window of optimal treatment initiation–though it is important to point out that a narrow window for optimal treatment response does rule out some treatment response outside that window. The development of tailored biomarkers for immune status may help inform antiviral drug scheduling.

The prediction of an optimal timing for antiviral therapy, as demonstrated in our simulation results, is indeed a fascinating and potentially crucial finding [15, 33]. While the existence of viral rebound is now firmly established in the clinical literature (see, for example Anderson et. al. [14] and the typical time course is broadly consistent with our simulations, we must emphasize that issue of optimal timing of therapy and its mechanistic basis is a novel prediction of the model and not yet clinically verified.

## Clinical relevance

Our mathematical model enables the multivariate exploration of various factors which can impact the clinical presentation and optimal treatment strategy for COVID-19. It allows for the simulation of simultaneous variations in patient comorbidities, antiviral effectiveness and enhanced immune escape by viral variants. A particualr advantage of a more comprehensive model such as ours is the ability to predict markers of the various model trajectories [15] and to develop hypotheses on predictors of response to therapy. Such markers could help locate an individual patient in the complex landscape of critical illness and suggest which variables, or 'treatable traits'[33] (e.g., viral replication, cytokine signaling, neutrophil recruitment etc.) should be targeted to achieve optimum results in a given setting. Such predictions could ultimately assist with predictive enrichment schemes and lead to more rationally designed clinical trials. In addition, as done here with antiviral therapy, it is possible to explore the range of outcomes that follow from other types of treatments such as anticoagulants and immunomodulation. Such studies could ultimately make sense of heterogenous clinical trial data [15].

In our manuscript, we detailed the model's potential for predicting future events related to COVID-19 and outlined the validation procedure. The model's predictive capability is built on its comprehensive representation of viral dynamics and immune responses. It can project the course of the infection under varying scenarios, such as changes in viral strains or treatment

strategies. For validation, we utilized a multi-step approach. Initially, the model was calibrated with current epidemiological data and clinical findings. Subsequently, its predictions were cross-verified with independent datasets, including emerging data on new viral variants and treatment responses. This ongoing validation process ensures the model remains accurate and relevant, providing valuable foresight in managing the pandemic.

## Supporting information

**S1 Appendix. Supplementary text, Figs and Tables.**
(DOCX)

## Author Contributions

**Conceptualization:** Mohammad R. Nikmaneshi.

**Funding acquisition:** Chrysovalantis Voutouri.

**Investigation:** Chrysovalantis Voutouri.

**Methodology:** Chrysovalantis Voutouri, C. Corey Hardin, Vivek Naranbhai, Mohammad R. Nikmaneshi, Melin J. Khandekar, Lance L. Munn, Triantafyllos Stylianopoulos.

**Project administration:** Lance L. Munn, Rakesh K. Jain.

**Software:** Chrysovalantis Voutouri.

**Supervision:** Lance L. Munn, Rakesh K. Jain, Triantafyllos Stylianopoulos.

**Writing – original draft:** Chrysovalantis Voutouri, C. Corey Hardin, Lance L. Munn, Rakesh K. Jain, Triantafyllos Stylianopoulos.

**Writing – review & editing:** Vivek Naranbhai, Mohammad R. Nikmaneshi, Melin J. Khandekar, Justin F. Gainor, Rakesh K. Jain, Triantafyllos Stylianopoulos.

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
