## [Decision Letter · Decision Letter 0]

9 Jan 2024

PONE-D-23-37489Dynamic heterogeneity in COVID-19: Insights from a mathematical modelPLOS ONE

Dear Dr. Stylianopoulos,

Thank you for submitting your manuscript to PLOS ONE. After careful consideration, we feel that it has merit but does not fully meet PLOS ONE’s publication criteria as it currently stands. Therefore, we invite you to submit a revised version of the manuscript that addresses the points raised during the review process.

Your manuscript has been deeply evaluated by two expert reviewers in the field of mathematical models: both found merit  in the study but  at the same time they raised many points to be considered, and I completely agree with their indications.

We look forward to receiving your revised manuscript.

Kind regards,

Vittorio Sambri, M.D., Ph.D.

Academic Editor

PLOS ONE

 [This work was supported by Rakesh Jain’s research is supported by R01-CA259253, R01-CA208205, R01-NS118929, U01-CA261842, and U01-CA 224348, Outstanding Investigator Award R35-CA197743 and grants from the National Foundation for Cancer Research, Jane’s Trust Foundation, Niles Albright Research Foundation and Harvard Ludwig Cancer Center. Lance Munn’s research is supported by R01-CA2044949. Triantafyllos Stylianopoulos’s research is supported by the European Research Council ERC-2019-CoG-863955. Chrysovalantis Voutouri is supported by Marie Skłodowska Curie Actions Individual Fellowship Global Horizon 2020 MSCA-IF-GF-2020-101028945.].  

[This work was supported by Rakesh Jain’s research is supported by R01-CA259253, R01-CA208205, R01-NS118929, U01-CA261842, and U01-CA 224348, Outstanding Investigator Award R35-CA197743 and grants from the National Foundation for Cancer Research, Jane’s Trust Foundation, Niles Albright Research Foundation and Harvard Ludwig Cancer Center. Lance Munn’s research is supported by R01-CA2044949. Triantafyllos Stylianopoulos’s research is supported by the European Research Council ERC-2019-CoG-863955. Chrysovalantis Voutouri is supported by Marie Skłodowska Curie Actions Individual Fellowship Global Horizon 2020 MSCA-IF-GF-2020-101028945.]

 [This work was supported by Rakesh Jain’s research is supported by R01-CA259253, R01-CA208205, R01-NS118929, U01-CA261842, and U01-CA 224348, Outstanding Investigator Award R35-CA197743 and grants from the National Foundation for Cancer Research, Jane’s Trust Foundation, Niles Albright Research Foundation and Harvard Ludwig Cancer Center. Lance Munn’s research is supported by R01-CA2044949. Triantafyllos Stylianopoulos’s research is supported by the European Research Council ERC-2019-CoG-863955. Chrysovalantis Voutouri is supported by Marie Skłodowska Curie Actions Individual Fellowship Global Horizon 2020 MSCA-IF-GF-2020-101028945.]

7. Thank you for stating the following in the Competing Interests section: 

[Conflict of Interest: JFG has served as a compensated consultant or received honoraria from Bristol-Myers Squibb, Genentech/Roche, Ariad/Takeda, Loxo/Lilly, Blueprint, Oncorus, Regeneron, Gilead, Moderna, Mirati, AstraZeneca, Pfizer, Novartis, iTeos, Nuvalent, Karyopharm, Beigene, Silverback Therapeutics, Merck, and GlydeBio; research support from Novartis, Genentech/Roche, and Ariad/Takeda; institutional research support from Bristol-Myers Squibb, Tesaro, Moderna, Blueprint, Jounce, Array Biopharma, Merck, Adaptimmune, Novartis, and Alexo; and has an immediate family member who is an employee with equity at Ironwood Pharmaceuticals. LLM owns equity in Bayer AG and is a consultant for SimBiosys. RKJ received consultant fees from Elpis, Innocoll, SPARC, SynDevRx; owns equity in Accurius, Enlight, Ophthotech, SynDevRx; and serves on the Boards of Trustees of Tekla Healthcare Investors, Tekla Life Sciences Investors, Tekla Healthcare Opportunities Fund, Tekla World Healthcare Fund; and received a grant from Boehringer Ingelheim. Neither any reagent nor any funding from these organizations was used in this study. Other co-authors have no conflict of interests to declare]. 

8. PLOS requires an ORCID iD for the corresponding author in Editorial Manager on papers submitted after December 6th, 2016. Please ensure that you have an ORCID iD and that it is validated in Editorial Manager. To do this, go to ‘Update my Information’ (in the upper left-hand corner of the main menu), and click on the Fetch/Validate link next to the ORCID field. This will take you to the ORCID site and allow you to create a new iD or authenticate a pre-existing iD in Editorial Manager. Please see the following video for instructions on linking an ORCID iD to your Editorial Manager account: " ext-link-type="uri" xlink:type="simple">https://www.youtube.com/watch?v=_xcclfuvtxQ".

Reviewers' comments:

Reviewer's Responses to Questions

**Comments to the Author**

1. Is the manuscript technically sound, and do the data support the conclusions?

Reviewer #1: Yes

Reviewer #2: Yes

2. Has the statistical analysis been performed appropriately and rigorously? 

Reviewer #1: Yes

Reviewer #2: N/A

3. Have the authors made all data underlying the findings in their manuscript fully available?

Reviewer #1: No

Reviewer #2: Yes

4. Is the manuscript presented in an intelligible fashion and written in standard English?

Reviewer #1: Yes

Reviewer #2: Yes

5. Review Comments to the Author

Reviewer #1: This manuscript by Voutouri and coworkers is an interesting work trying to translate the complexity of the interactions between a virus (SARS-CoV-2), its host and possible countermeasures (vaccines, antivirals) in mathematical terms. The adopted equations are sound and took into account several parameters. The conclusions are relevant. The manuscript is surely worth of publication. However, some aspects should be addressed before publication as detailed below:

1. What this Referee is missing is the actual application (if any) of the mathematical model developed by the Authors to Public Health. What do the Authors envision? Please explain better in the Discussion. This is relevant especially for Readers of PlosOne not entirely familiar or interested to mathematical models but more into medical/biological Research

2. I would clearly separate the Results in those coming out from the model in line with what is already well established in the COVID-19 field and supported by well controlled studies, those suggesting novel findings that need to be further addressed with specific experiments, and those likely contradicting published data. All these aspects may be represented in a Table or diagram and more than one reference should be reported for any result supporting already published data. In my opinion, this would make the manuscript clearer and more interesting for readers less familiar with mathematical models and would increase the overall merit of the study

3. In Figure 1 I would change the title as what it is shown is not the mathematical model, but the biological interplay took into account by the Authors to implement their model. Furthermore, I would use part of what is now reported in the Description of the Mathematical Model in the Supplementary file to better explain what is depicted in the Figure. Also the Authors should clearly state in the main text why they selected those pathways and based on what literature data (with references). This is essential as it constitutes the basis of the entire work

4. For a clear understanding of Figure 2, the Authors should explain what are the variants they took into account, what do they mean by favorable and unfavorable and form what published data they come to this latter definition

5. In Supplementary Table 1, every time the Authors adopt an estimate of the relative parameter from where this estimate comes from? This is very important and I could not find this information

6. Supplementary Table 2 is not clear to me? From where the Kin comes from? It is calculated? If yes from which one of the equations? If is estimated, from where? This is another crucial aspect of the manuscript. Please explain

Reviewer #2: Referee report on the paper Dynamic heterogeneity in COVID-19: Insights from a mathematical model

(PONE-D-23-37489)

The main result of the paper is to prove the heterogeneity of the clinical courses of COVID-19 as a consequence of random factors like the innate or adaptive immune response and the initial viral load by using the simulation of a mechanistic mathematical model of COVID-19 that have been developed by the authors.

The referee is not an expert in the clinical treatments the COVID-19 disease so that my comment refers to the model properties.

The model is only schematically described in the paper since it has been presented in previous papers, but the equations are reported in the supplementary material together with an impressive list of parameters that define the model and some of them are only estimated so that the existence of an overfitting problem is possible using experimental data to validate the model.

The model could be interpreted as a proposal of a`digital twin' for the COVID-19 evolution in a patient,

to perform in silico experiments, but the validation problem is still open and should be better discussed in the paper.

It is the referee's opinion that the utility of any dynamical model is the capability of providing predictions on the future evolution of the considered phenomenon using the available information at a given time and a validation procedure is realized when the model is able to predict situations not yet observed. It would be useful for the reader to know this type of validation procedure has been considered or if the paper is the first attempt to propose a validation procedure of the model.

The idea of the author to consider a range of values for the parameters is the usual procedure to estimate the sensitivity of the model to the parameter changes and to highlight the relevant parameters. Considering the complexity of the model it is not clear in the paper if there exists some control parameter whose values is critical to understand the model simulations, whereas the other parameter values have a more limited effect. Which are the control parameters of the model? The simulations presented in the paper have been chosen to test the existence of control parameters?

The authors proposed three sets of simulations to study:

the interaction of antiviral therapy with heterogeneity in vaccine induced immunity;

the interaction of antiviral therapy with heterogeneity in the innate and adaptive immune response;

the interaction of initial viral load with treatment efficacy and immune response.

In each simulation a parameter value is varied by order of magnitudes, but there is not a clear explanation if such a variability of the values are consistent with observed data or can be justified from a physiological point of view.

The results of the simulations are not unexpected, so that it is not clear if the complexity of the model is really justified: can the authors comment on the need of such a complex model to get the results presented in the paper?

Moreover it is not clear to me how to quantify the color scale in the right of the figures that distinguishes the cases between favorable and unfavorable. It would be useful if the author could give more information on how the simulation results are classified.

The last simulation in the paper considers the `rebound after antiviral therapy and optimal treatment initiation': this is an interesting case since the results show the existence of an optimal timing for the antiviral therapy. However the question if this effect is really observed in the available data is not clear, may the authors comment if this prediction is consistent with some observations.

In the final discussion the authors stress that `a major finding of the paper is the confirmation that patient clinical course may be heavily influenced by random events which are difficult to observe or control for in clinical cohorts', but this is really an unexpected result that characterizes the COVID-19 disease and justifies the complexity of the model? Due to the large variation in the parameter values, one would expect a variability in the simulation results for any mathematical model of a disease evolution, why this result is relevant for the proposed model?

The authors claim that an advantage of the model is to predict markers of the various model trajectories in order to optimize the therapy results. This is a key point for the model, but there is no suggestion of possible predictors in the paper. To define a predictor is difficult due to the lack of the necessary data from the patients, or due to the complexity of the model?

In summary, I would like to recommend the paper for publication in PLOS One, once the authors have replied to previous comments.

6. PLOS authors have the option to publish the peer review history of their article (what does this mean?). If published, this will include your full peer review and any attached files.

Reviewer #1: No

Reviewer #2: **Yes: **Armando Bazzani

---

## [Author Response · Author response to Decision Letter 0]

22 Jan 2024

Response to Reviewers

We thank both reviewers for their constructive comments and suggestions, which have helped us revise and strengthen our manuscript. 

Please find our point-by-point response (in italics) to both reviewers’ comments below. We have marked edits in the revised manuscript and supplementary files in red font. 

Reviewer #1

This manuscript by Voutouri and coworkers is an interesting work trying to translate the complexity of the interactions between a virus (SARS-CoV-2), its host and possible countermeasures (vaccines, antivirals) in mathematical terms. The adopted equations are sound and took into account several parameters. The conclusions are relevant. The manuscript is surely worthy of publication. However, some aspects should be addressed before publication as detailed below: 

We thank the Reviewer for the positive evaluation of our work.

1. What this Referee is missing is the actual application (if any) of the mathematical model developed by the Authors to Public Health. What do the Authors envision? Please explain better in the Discussion. This is relevant especially for Readers of PlosOne not entirely familiar or interested to mathematical models but more into medical/biological Research.

We thank the Reviewer for the valuable feedback. We have expanded the Discussion section to elucidate the practical applications of our mathematical model in public health. Specifically, we emphasize that the complexity highlighted by our model requires a more nuanced interpretation of clinical trial data – in particular since initial conditions such as size of viral inoculum are not always observable, this complexity is a challenge to the external validity of trial cohorts. We have provided examples and scenarios demonstrating how our model can aid in decision-making and policy formulation in the context of COVID-19 management. (page 17)

“Our mathematical model enables the multivariate exploration of various factors which can impact the clinical presentation and optimal treatment strategy for COVID-19. It allows for the simulation of simultaneous variations in patient comorbidities, antiviral effectiveness and enhanced immune escape by viral variants.” 

By forecasting the outcomes of different treatment approaches, the model can identify the most promising strategies, improving the overall response to COVID 19 outbreaks. 

In summary, our model's ability to simulate and analyze diverse clinical scenarios positions it as a key tool to generate and explore hypothesis for later exploration in clinical trials. Its applications range from formulating optimal treatment strategies for diverse patients to predicting inclusion and exclusion criteria to address heterogeneity optimize power in future clinical trials.

2. I would clearly separate the Results in those coming out from the model in line with what is already well established in the COVID-19 field and supported by well controlled studies, those suggesting novel findings that need to be further addressed with specific experiments, and those likely contradicting published data. All these aspects may be represented in a Table or diagram and more than one reference should be reported for any result supporting already published data. In my opinion, this would make the manuscript clearer and more interesting for readers less familiar with mathematical models and would increase the overall merit of the study.

We appreciate your suggestion to categorize the results. In the revised manuscript, we included a Supplementary table in the Results section, delineating the findings into three distinct categories: those consistent with established COVID-19 knowledge, novel findings requiring clinical validation, and findings that potentially contradict existing data. We have also provided multiple references for each category to enhance the clarity and utility of the manuscript.

Model predictions consistent with existing data Novel model predictions requiring clinical validation Model predictions potentially contradicting existing data

Viral neutralization by vaccine induced antibodies in setting of infection with omicron variants from BA.1 to BA 4/5 [1-4]

Our results confirm that antiviral therapy confers benefit even in the setting of vaccination and that effective antiviral therapy is more important to outcome in the setting of variants which evade vaccine induced immunity. [5]

We find that when pro-inflammatory cytokine production is increased, the decreased viral load comes at the cost of poorer gas exchange. Increasing the effectiveness of anti-viral therapy in the setting of increased inflammatory cytokine production attenuates the adverse effects on gas exchange (consistent with the more rapid decrease in inflammatory stimulus). [6, 7]

The proportion of patients that recovered from COVID-19 having been treated with antivirals, remdesivir, molnupiravir and nirmatrelvir+ritonavir (NMV/r) [8-10]

Effective antiviral therapy is more important to treatment outcome in the setting of variants which evade vaccine induced immunity[11, 12]. 

When neutrophil population is set to zero, the model predicts worsening oxygen saturation with elevated levels of cytokine production, as in the case with neutrophils present[13-15]

Viral load rebound after antiviral therapy [16]

High rates of immune cell activation and inflammatory cytokine production result in lower viral load – and that viral control is augmented by effective antiviral therapy[17-19]

Very high initial viral loads can result in high peak viral loads even at very high levels of expression of proinflammatory cytokines

[20-22]

Supplementary Table 5. Categorization of Model Findings in COVID-19 Research. The table delineates the findings derived from our mathematical model into three categories: (1) Model predictions consistent with existing data, (2) Novel model predictions requiring clinical validation, and (3) Model predictions potentially contradicting existing data.

3. In Figure 1 I would change the title as what it is shown is not the mathematical model, but the biological interplay taken into account by the Authors to implement their model. Furthermore, I would use part of what is now reported in the Description of the Mathematical Model in the Supplementary file to better explain what is depicted in the Figure. Also the Authors should clearly state in the main text why they selected those pathways and based on what literature data (with references). This is essential as it constitutes the basis of the entire work.

Thank you for pointing this out. We have revised the title of Figure 1 to more accurately reflect its content “Figure 1: Biological interplay and pathways incorporated by the mathematical model.”, focusing on the biological interplay considered in our model. Additionally, we have incorporated relevant details from the supplementary file into the main text to provide a clearer explanation of the figure.

4. For a clear understanding of Figure 2, the Authors should explain what are the variants they took into account, what do they mean by favorable and unfavorable and form what published data they come to this latter definition.

We acknowledge the need for clarity regarding the variants considered (all omicron variants from BA.1 to BA 4/5) and the definitions of 'favorable' and 'unfavorable' outcomes. 'Favorable' outcomes for viral load correspond to low concentrations, while 'unfavorable' outcomes relate to high concentrations for viral load. For oxygen saturation, 'favorable' outcomes are indicated by high levels, and 'unfavorable' outcomes by low levels oxygen saturation. We have updated the manuscript to include a detailed explanation of these terms and their relevance to our model in both the figure description and the main text.

5. In Supplementary Table 1, every time the Authors adopt an estimate of the relative parameter from where this estimate comes from? This is very important and I could not find this information

We realize the importance of transparency regarding our parameter estimates. We estimated these parameters during the validation of the model. This approach was crucial to ensure that our model accurately reflects the dynamics of COVID-19 infection and response to treatments. To make this clear, we replaced “estimated” with “from model validation”.

6. Supplementary Table 2 is not clear to me? From where the Kin comes from? It is calculated? If yes from which one of the equations? If is estimated, from where? This is another crucial aspect of the manuscript. Please explain

Thank you for pointing out the need for clarity regarding the Kin parameter in Supplementary Table 2. The Kin parameter, representing the rate of release of replicated virus, is indeed a calculated value, determined through the validation process of the model for each antiviral drug. This calculation is integral to the model's ability to accurately simulate the effects of antiviral treatments on the virus dynamics within the body. Kin is calibrated to reproduce clinical data on the effect of antiviral therapy on viral load. 

Reviewer #2

The main result of the paper is to prove the heterogeneity of the clinical courses of COVID-19 as a consequence of random factors like the innate or adaptive immune response and the initial viral load by using the simulation of a mechanistic mathematical model of COVID-19 that have been developed by the authors.

The referee is not an expert in the clinical treatments the COVID-19 disease so that my comment refers to the model properties.

1. The model is only schematically described in the paper since it has been presented in previous papers, but the equations are reported in the supplementary material together with an impressive list of parameters that define the model and some of them are only estimated so that the existence of an overfitting problem is possible using experimental data to validate the model. 

The model could be interpreted as a proposal of a`digital twin' for the COVID-19 evolution in a patient, to perform in silico experiments, but the validation problem is still open and should be better discussed in the paper.

We appreciate your insight regarding the description of our model and the potential issue of overfitting. In response, we have expanded the model description in the paper and addressed the concern of overfitting, particularly by discussing the methods employed to mitigate this risk. (page 4)

“This model encompasses intricate interactions occurring within the human body, utilizing differential equations to simulate various aspects of SARS-CoV-2 infection and the corresponding immune responses. These include the viral entry process, immune system activation, cytokine production, and the coagulation cascade. Furthermore, our model incorporates mechanisms and immune responses related to both mRNA and vector-based vaccines. Of particular importance is the inclusion of a pharmacokinetic-pharmacodynamic model, which meticulously tracks the movement of viral particles and other relevant elements across major body compartments. This comprehensive framework empowers us to conduct in-depth analyses and make predictions.”

2. It is the referee's opinion that the utility of any dynamical model is the capability of providing predictions on the future evolution of the considered phenomenon using the available information at a given time and a validation procedure is realized when the model is able to predict situations not yet observed. It would be useful for the reader to know this type of validation procedure has been considered or if the paper is the first attempt to propose a validation procedure of the model.

Your point on model validation is well-taken. We have included a section in the Discussion where we elaborate on the model's potential for predicting future events and detail the proposed validation procedure. (page 18)

“In our manuscript, we detailed the model's potential for predicting future events related to COVID-19 and outlined the validation procedure. The model's predictive capability is built on its comprehensive representation of viral dynamics and immune responses. It can project the course of the infection under varying scenarios, such as changes in viral strains or treatment strategies. For validation, we utilized a multi-step approach. Initially, the model was calibrated with current epidemiological data and clinical findings. Subsequently, its predictions were cross-verified with independent datasets, including emerging data on new viral variants and treatment responses. This ongoing validation process ensures the model remains accurate and relevant, providing valuable foresight in managing the pandemic.” 

We would also like to highlight an additional utility of a mechanistic model such as ours. As the reviewer correctly points outs, the large number of parameters increases the danger of overfitting and may complicate the use of the model as a ‘digital twin,’ able to predict the clinical course of future individual patients. However, our approach enables the exploration of a large number of pathophysiologic scenarios and therefore the development of hypotheses about the sources of heterogeneity in COVID-19 clinical courses. While this may seem a less compelling goal than the prediction of individual patient outcomes, we would like to point out that such heterogeneity is a major and much discussed challenge in critical care clinical research (see for example,Leligdowicz, NEJM Evid 2022;1(11) and Maslove et. al. Nature Medicine volume 28, pages1141–1148 (2022)). The sources of such heterogeneity are currently poorly understood but can be explored with a mechanistic model. As an example, the reviewer has highlighted that a major finding of this work is to prove the heterogeneity of the clinical courses of COVID-19 as a consequence of random factors like the innate or adaptive immune response and the initial viral load. Such random or dynamic sources of heterogeneity are rarely, if ever, discussed in the clinical critical care literature. 

3. The idea of the author to consider a range of values for the parameters is the usual procedure to estimate the sensitivity of the model to the parameter changes and to highlight the relevant parameters. Considering the complexity of the model it is not clear in the paper if there exists some control parameter whose values is critical to understand the model simulations, whereas the other parameter values have a more limited effect. Which are the control parameters of the model? The simulations presented in the paper have been chosen to test the existence of control parameters?

The Reviewer is right that the identification of control parameters is crucial. We have now pinpointed and discussed the key control parameters in the Results section, elucidating their impact on the model simulations. (page 7-8)

“In our study, key control parameters that impact COVID-19 outcomes include the basic reproduction number, antiviral treatment efficacy, immune response rates, and initial viral load, which all influence epidemic peaks and virus spread. Variations in treatment efficacy markedly affect disease progression and population spread dynamics. Differing immune responses lead to varied disease severities and recovery rates, highlighting individual immune variability. The initial viral load also plays a vital role, with higher loads linked to more severe disease courses. Understanding these parameters is essential for applying our model to real-world scenarios.”

4. The authors proposed three sets of simulations to study:

the interaction of antiviral therapy with heterogeneity in vaccine induced immunity;

the interaction of antiviral therapy with heterogeneity in the innate and adaptive immune response;

the interaction of initial viral load with treatment efficacy and immune response.

In each simulation a parameter value is varied by order of magnitudes, but there is not a clear explanation if such a variability of the values are consistent with observed data or can be justified from a physiological point of view.

We acknowledge the necessity of justifying parameter variability. In the revised manuscript, we have provided a rationale, grounded in physiological and experimental evidence, for th

---

## [Decision Letter · Decision Letter 1]

22 Mar 2024

Dynamic heterogeneity in COVID-19: Insights from a mathematical model

PONE-D-23-37489R1

Dear Dr. Stylianopoulos,

We’re pleased to inform you that your manuscript has been judged scientifically suitable for publication and will be formally accepted for publication once it meets all outstanding technical requirements.

Kind regards,

Yury E Khudyakov, PhD

Academic Editor

PLOS ONE

Additional Editor Comments (optional):

Reviewers' comments:

Reviewer's Responses to Questions

**Comments to the Author**

1. If the authors have adequately addressed your comments raised in a previous round of review and you feel that this manuscript is now acceptable for publication, you may indicate that here to bypass the “Comments to the Author” section, enter your conflict of interest statement in the “Confidential to Editor” section, and submit your "Accept" recommendation.

Reviewer #1: All comments have been addressed

2. Is the manuscript technically sound, and do the data support the conclusions?

Reviewer #1: Yes

3. Has the statistical analysis been performed appropriately and rigorously? 

Reviewer #1: Yes

4. Have the authors made all data underlying the findings in their manuscript fully available?

Reviewer #1: Yes

5. Is the manuscript presented in an intelligible fashion and written in standard English?

Reviewer #1: Yes

6. Review Comments to the Author

Reviewer #1: The Authors have took into account my suggestions and addressed all the issues. The paper is now suitable for publication.

7. PLOS authors have the option to publish the peer review history of their article (what does this mean?). If published, this will include your full peer review and any attached files.

Reviewer #1: No

---

## [Editor Report · Acceptance letter]

29 Apr 2024

PONE-D-23-37489R1 

PLOS ONE

Dear Dr. Stylianopoulos, 

I'm pleased to inform you that your manuscript has been deemed suitable for publication in PLOS ONE. Congratulations! Your manuscript is now being handed over to our production team.

Kind regards, 

on behalf of

Dr. Yury E Khudyakov 

Academic Editor

PLOS ONE